# Challenges to and Strategies for the Climate Village Program Plus: A Lesson Learned from Indonesia

**Catur Budi Wiati** [1,*], **I Wayan Susi Dharmawan** [1], **Niken Sakuntaladewi** [1], **Sulistya Ekawati** [1], **Tien Wahyuni** [1], **Rizki Maharani** [1], **Yayan Hadiyan** [1], **Yosua Naibaho** [2], **Wahyudi Iman Satria** [2], **Ngatiman Ngatiman** [1], **Abdurachman Abdurachman** [1], **Karmilasanti Karmilasanti** [1], **Aulia Nur Laksmita** [1], **Eddy Mangopo Angi** [3] **and Chiranjeewee Khadka** [4,*]

1    National Research and Innovation Agency of Indonesia, Jakarta Pusat 10340, Indonesia; iway028@brin.go.id (I.W.S.D.); niken.sakuntaladewi@brin.go.id (N.S.); sulistya.ekawati@yahoo.co.id (S.E.); tien.wahyuni@brin.go.id (T.W.); rizki.maharani@brin.go.id (R.M.); yayan.hadiyan@brin.go.id (Y.H.); ngatimanforester@gmail.com (N.N.); abdurachman.1@brin.go.id (A.A.); karmilasanti@brin.go.id (K.K.); auli013@brin.go.id (A.N.L.)
2    Regional Council on Climate Change of East Kalimantan, Samarinda 75123, Indonesia; yosuanaibaho@gmail.com (Y.N.); wahyudiimansatria30@gmail.com (W.I.S.)
3    Independent Consultant for Forest and Governance, Samarinda 75119, Indonesia; eddymangopo@gmail.com
4    Global Change Research Institute (Czechglobe), Brno 603 00, Czech Republic
*    Correspondence: catur.budi.wiati@brin.go.id (C.B.W.); khadka.c@czechglobe.cz (C.K.)

**Abstract:** The Climate Village Program (CVP) is one of the national flagship programs of the Ministry of Environment and Forestry of the Republic of Indonesia to support emission reduction and climate resilience. This paper examines the challenges and strategies for implementing the climate village program in the national and sub-national contexts. Data and information derived from discussions, seminars, focus group discussions, and interviews with local government officials in East Kalimantan were used to analyze the social learning of the CVP plus, including those on the policy process and its concept, integration program, and implementation. Sustainable strategies need to be addressed by integrating the CVP plus into the medium-term development plan of the region. The challenges and way forward of the CVP plus could be an excellent lesson for implementation in all provinces of Indonesia to support FOLU (Forest Other Land Use) Net Sinker 2030 and LTS-LCCR (Long-Term Strategy on Low Carbon and Climate Resilience) 2050. Key challenges and strategies for the CVP plus are highlighted in the planning and implementation phases, especially in improving climate resilience. This study also points out the steps of implementation of the CVP, development partners and their roles in relation to climate change and other socio-economic facts that make it difficult to engage real stakeholders in the implementation of the CVP plus.

**Keywords:** Climate Village Program plus; emission reduction; forest; policy; socio-economic

## 1. Introduction

As an archipelagic country, Indonesia is very vulnerable to climate change. The Long-Term Strategy on Low Carbon and Climate Resilience (LTS-LCCR) 2050 has been developed to support integrated national transparency on climate change adaptation and mitigation actions. LTS-LCCR 205 covers 83,820 villages, of which 3270 villages have registered for the Climate Village Program (CVP) by 2021. The number of these CVPs is expected to increase gradually to 20,000 by 2024 [1].

The CVP is one of the community-based national flagship programs of the Ministry of Environment and Forestry (MoEF) of the Republic of Indonesia, which has been running since 2016. It aims to mainstream the global issue of climate change to collectively respond to climate change impacts occurring at the local site level. MoEF has created a roadmap as a common reference for implementing the climate village program. The CVP is followed and

implemented by sub-national parties to support emission reduction and climate resilience. Like other ambitious national policies, the CVP is dynamically implemented by provincial governments during the implementation process towards emission reduction and climate resilience, based on the variations of problems, potentials, and supports they have. In 2021, the guidelines and direction for the implementation of the CVP were issued by the MoEF Directorate of Climate Change [2]. An important note on the CPV is that information on forest land availability and carbon potential in villages was less important. However, as a joint adaptation and mitigation action, this program makes a significant contribution to GHG mitigation efforts in Indonesia before and after 2020 [3].

East Kalimantan is one of 34 provinces in the country threatened by deforestation and forest degradation. It is estimated that between 2003 and 2013, 230,720 ha of forest area was lost, and about 305 million tons of $CO_2$ will be released from protected areas if deforestation continues [4]. Important lessons have been learned in this province, including improving programs to combat deforestation and forest degradation. The result is an increase in the CVP, which is designed to maintain and/or improve forest cover.

The CVP plus was launched with multilateral support from the Forest Carbon Partnership Facility-Carbon Fund (FCPF-CF). East Kalimantan has an area of 12.7 million ha, of which 6.5 million ha (54%) is forested and has been selected as a site for pilot project FCPF-CF since 2015 [5].

The CVP plus is to operate in cultivation and protection areas, which are spread over five districts and one city. The CVP plus also plays a role in three sectors, namely forestry, plantation, management, and community empowerment. Efforts must be made to increase community resilience to address the impact of climate change [6,7]. The social learning process becomes an important component in the CVP plus to promote the active participation of the community and stakeholders in the implementation of local actions. This will strengthen resilience to the impacts of greenhouse gas (GHG) emissions [8]. Social learning defines a shift from "multiple cognition" to "collective cognition" as the condition of quite different perceptions, the potential for change, shared perspectives, insights, and values [9]. Social learning is understood as the process through which groups of people learn to define common problems, find, search for, and implement alternative options or solutions, and assess the value of solutions for specific problems [10,11].

As stated in [12], the social learning process is an approach to community development and education. It aims primarily to aid participation and experience in capacity building. This program of the CVP is superior to others because it is easy to implement, locally relevant, culturally appropriate, and efficient in its use of natural resources. There is an opportunity to receive incentives, although budgets are already available at the provincial, district, and village levels from the Indonesian central government.

This paper aims to assess the challenges and strategies in climate program implementation at the national and sub-national levels to support emission reduction and climate resilience, based on the experience of the CVP plus in East Kalimantan. The major gap is how to implement the CVP plus in line with local development plans and other climate change programs with stakeholders, as envisioned in the regional medium-term development plan. Another gap is how to implement the CVP plus at the local level in accordance with the characteristics of each village to reduce climate-related disaster risks at the local level, accommodate local wisdom as a strategy to stimulate communities to make efforts to increase their resilience to climate change, and protect tropical rainforest through community empowerment and participation programs that are more collaborative and well-targeted.

## 2. Materials and Methods

### 2.1. Research Framework

This research is a case study focused on the efforts of various parties in East Kalimantan Province to support villages with forest cover and carbon stock to gain result-based payment (RBP) from deforestation and forest degradation prevention efforts. East Kali-

mantan Province was selected as the research site for a number of reasons. First, local communities in East Kalimantan have the ability to mitigate and adapt to climate change. Second, the government and development partners appreciate and support the actions of local communities in mitigating climate change. Third, East Kalimantan is the site of the first pilot project for the jurisdiction-based reduction of emissions from deforestation and forest degradation (REDD+) in Indonesia. The REDD+ pilot project is in the FCPF-CF scheme. The logical framework of this research is provided in Figure 1.

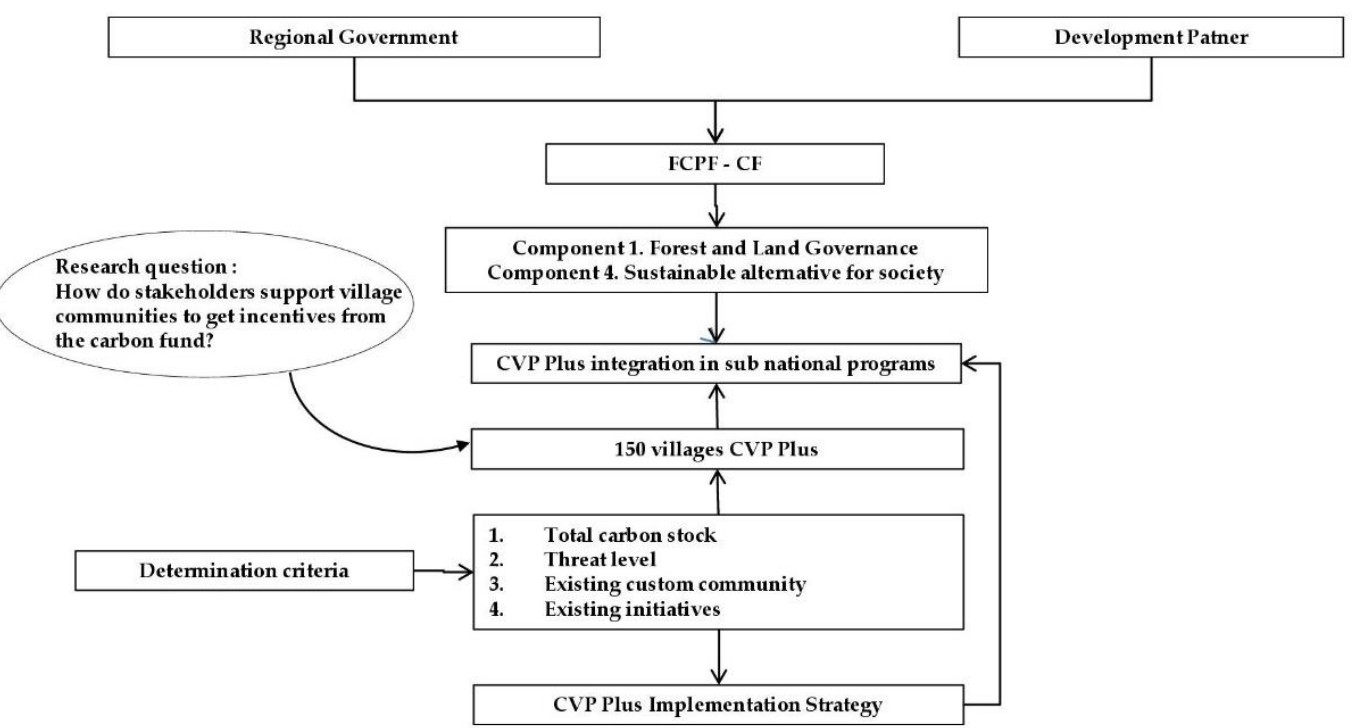

**Figure 1.** Research logical framework.

### 2.2. Study Location

This study was conducted in the province of East Kalimantan. A total of 10 CVP plus villages were purposely selected, representing various districts, forest types, remaining forest, the presence of indigenous peoples/customary forests, the existence of forest Adat/Custom communities in the CVP plus villages [13,14], existing FPIC approval in accordance with FPIC guidelines [13,15–18], and the existence of mitigation and adaptation efforts to climate change for the sustainability of people's livelihoods. The selected villages are Angsa Slutung Village and Muara Andeh Village in Paser Regency, Bermai Village and Sembuan Village in West Kutai Regency, Senyiur Village and Melan Village in East Kutai Regency, as well as Felt Village, Long Lanuk Village, Punan Segah Village, and Village Pegat Batumbuk in Berau Regency. The map of study locations is shown in Figure 2.

### 2.3. Data Collection

A preliminary study was conducted from December 2018 to March 2019 to review CVP policies at the national level and CVP plus policies at the sub-national level and conduct interviews with key informants from local government and development partners. A follow-up study on the implementation of the CVP plus in the field was carried out from August 2020 to December 2021.

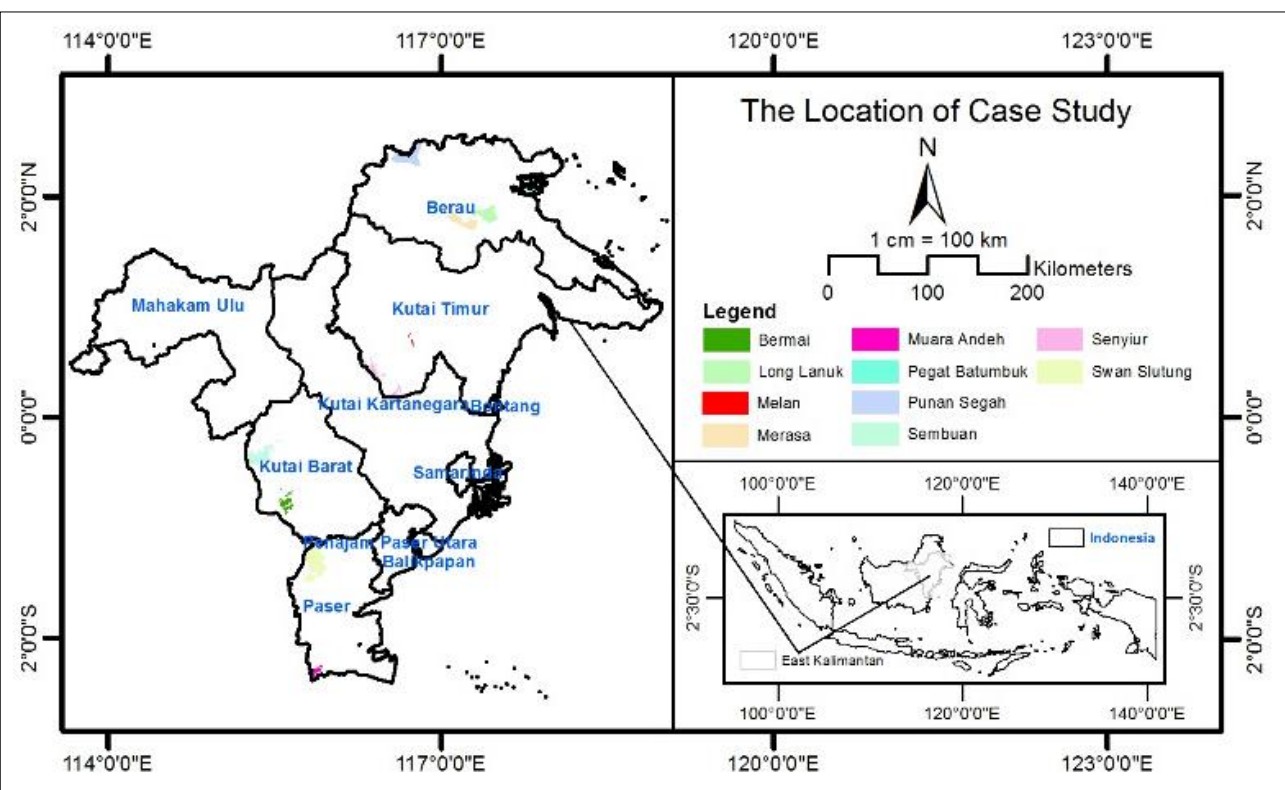

**Figure 2.** Map of study locations. Source: author's compilation.

Research data were obtained in the form of primary and secondary data from key informant interviews, focus group discussions (FGD), field observation, and desk studies. Primary data in the form of quantitative and qualitative data include forest cover area and potential carbon stock, forest resource utilization activities, and the presence of indigenous/customary communities. Secondary data collected include administrative data and data on the socio-economic conditions of rural communities, as well as documents on adaptation and mitigation of climate change in East Kalimantan.

Carbon stock was measured using area activity data and carbon stock of the forest cover [5]. The activity data were taken from the map of land cover in the form of forest in 2019. The forest carbon stock values of primary dryland forest (281.3 tC/ha), secondary dryland forest (147.3 tC/ha), primary mangrove forest (160.8 tC/ha), secondary mangrove forest (126.8 tC/ha), primary swamp forest (344.24 tC/ha), and secondary swamp forest (233.5 tC/ha). According to the [19], the sources of carbon stock consist of five carbon pools: above-ground biomass, below-ground biomass, dead wood, litter, and soil organic matter. The formula for estimating the carbon stock at landscape level ($CS_{landscape}$) for each forest cover ($A_i$) and carbon stock per hectare ($CS_i$) as follows $CS_{landscape} = A_i \times CS_i$.

### 2.4. Key Respondents

The key respondents were divided into three major groups. The first group was the group of the provincial government and local organizations of East Kalimantan Province, consisting of the Forestry Service of East Kalimantan Province, the Environment Service of East Kalimantan Province, the Community and Village Empowerment Service of East Kalimantan Province, the Plantation Service of East Kalimantan Province, the Energy and Mineral Resources Service of East Kalimantan Province, and the Regional Disaster Mitigation Agency of East Kalimantan Province. The second group was the group of development partners, consisting of the Provincial Climate Change Council (DDPI) of East Kalimantan, Yayasan Konservasi Alam Nusantara (YKAN), GIZ Forclime FC, WWF Indonesia, Planet Urgence, GGGI, BUMI Foundation, Yayasan Biosfer Manusia (BIOMA Foundation), PADI

Foundation, Yayasan Konservasi Khatulistiwa (YASIWA) Indonesia, ULIN Foundation, and Mulawarman Environment Forum (FliM). Meanwhile, the third group consisted of community groups in the 10 CVP villages, namely the Village Government, the Village Council (BPD), the Community Empowerment Agency (LPM), the Village Adat/Custom Institution, the Village Forest Management Agency (LPHD), the Adat/Custom Forest Management Agency (LPHA), and Karang Taruna.

### 2.5. Analysis Methods

An analysis was conducted using descriptive, qualitative, and quantitative methods by exposing the processes involved, from planning (determining the parties involved, determination of sites, budgeting, and creation of the FPIC team according to needs and the FPIC guide), to implementation (socialization and FPIC approval), and reporting (the CVP plus villages who declared their participation in the FCPF-CF program through the FPIC phases and were also actively involved in reporting the emission reduction activities on the MMR portal of East Kalimantan).

## 3. Results

### 3.1. The CVP plus and Support from Various Parties

#### 3.1.1. The Climate Village plus Policy and Program

The CVP is one of the national programs that demonstrate the Indonesian Government's commitment to the climate change mitigation efforts under the Paris Agreement. Indonesia adopted this agreement in 2014 at COP-12 as outlined in the Nationally Determined Contribution (NDC) document as part of the low-casualty development and climate change adaptation activities. As President Joko Widodo stated at the opening event of the Climate Adaptation Summit 2021, the Indonesian Government has set a goal to establish 20,000 climate villages by 2024. As of 2021, 3270 locations across Indonesia have been registered as CVP sites. This program is implemented in low administrative regions such as *rukun warga* (citizen association) or *dusun* (hamlet) and high administrative regions such as *kelurahan* or *desa* (village) or in an area whose community has made efforts to sustainably adapt to and mitigate climate change [20].

The CVP applies community-based development according to these three principles: community-based, local resource-based, and sustainable. The CVP plus or the Low-Emission CVP is one of the programs initiated by the Forest Carbon Partnership Facility (FCPF), which aims to integrate the CVP with the Carbon Fund (CF) scheme, particularly in villages with remaining forests. The FCPF implementation is a follow-up to land-based REDD+ payment preparation in Indonesia that takes the form of a result-based payment. This program rewards people in certain locations who have implemented climate change adaptation and mitigation efforts in a sustainable manner [20–22]. REDD+ is instrumental in engaging local communities and Adat/Custom community in the low emission program [23].

The villages targeted by the CVP plus under the FCPF program differ slightly from the villages targeted by the CVP implemented by the MoEF. The CPV plus takes into account criteria for carbon stock, the level of threat from forest destruction, the existence of Adat/Custom communities [24], existing CVP-related initiatives, and the urgency for these villages to become climate villages. Following the ERPD document, the East Kalimantan Provincial Government has identified approximately 150 CVP plus villages with remaining forest based on four criteria: carbon stock, levels of threat, the existence of Adat/Custom local communities, and existing support [1,14]. The CVP plus implementation process is illustrated in Figure 3.

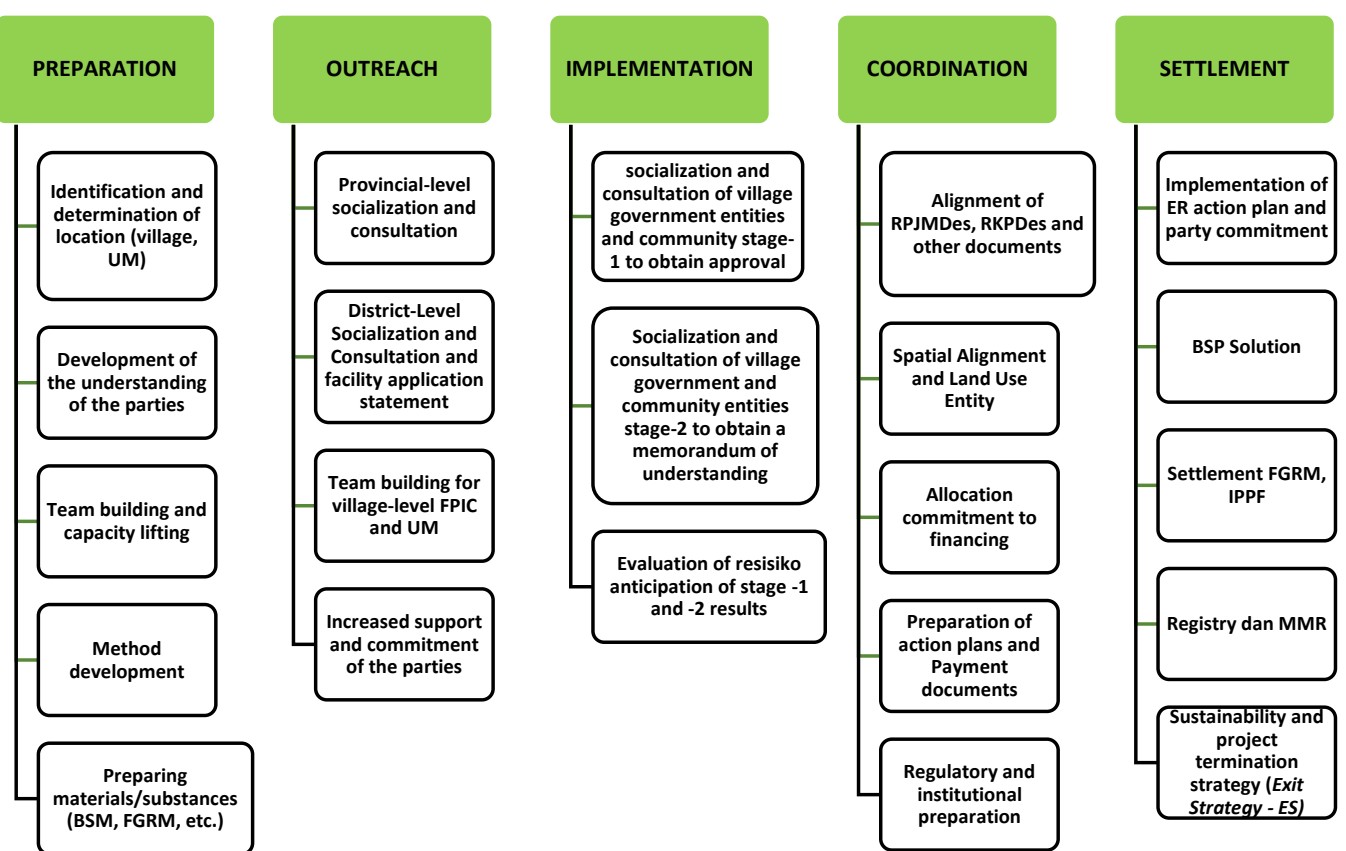

**Figure 3.** Steps of FPIC CVP plus implementation in East Kalimantan.

3.1.2. The Climate Change Mitigation Activities Conducted by the Local Government and the Development Partners in East Kalimantan

Environmental regulations are critical to encouraging better environmental practices [25]. The effectiveness of climate change policy implementation is not only measured by the right content of the policy but also affected by the role of stakeholders who implement the policy [26]. The stakeholders, including the regional heads, development partners, and private companies, also have critical roles in supporting climate change issues [19–21], including those located in East Kalimantan. They have taken a central role in the CVP's success. The term CVP plus is used as an alternative way of naming the program to accommodate the climate change adaptation and mitigation activities previously undertaken by various parties in East Kalimantan. The role of each stakeholder in climate change mitigation and adaptation in East Kalimantan is explained in Table 1.

**Table 1.** The role of local governments, development partners, and business entities in relation to climate change in East Kalimantan.

| Institution | Number of Institutions Involved | Program/Activity |
|---|---|---|
| The regional apparatus organizations (RAOs) | 9 | Determination of Climate Villages; Fire Awareness Community; develop five schemes of Social Forestry; determination of disaster-resilient villages; new and renewable energy; building Village Index Development; Fire Awareness Farmers Group |

**Table 1.** *Cont.*

| Institution | Number of Institutions Involved | Program/Activity |
|---|---|---|
| Development partner | 14 | Sustainable forest management; community conservation area; sustainable energy; rhinoceros habitat protection and carbon calculation; SIGAP Village; sustainable oil palm plantation development policy; peat social forestry and sustainable peat management and protection; Green growth planning, FCPF-CF support, institutional capacity improvement, planning, and business plan development; Conservation, protection, restoration, and sustainability of forest use; sustainable community oil palm plantation development; Social forestry, conflict management; budget transparency; Forest carbon management; Community training in relation to carbon and conflict resolution; Out-of-forest forest planning and management; Local institutional improvement, mangrove forest restoration, and local conservation; Mangrove forest restoration |
| Field of Business | 6 | Fire Awareness Community Farmers Group; Sustainable Palm Oil Estate under Indonesian Sustainable Palm Oil Certification (ISPO) and Roundtable on Sustainable Palm Oil (RSPO; High Conservation Value (HCV) plantation; High Conservation Value Forest (HCVF); Corporate Social Responsibility (CSR); Post-mining reclamation; Environment impact assessment (Analisis Dampak Lingkungan/AMDAL); environment management plan (Rencana Pengelolaan Lingkungan/RKL); environment monitoring plan (Rencana Pemantauan Lingkungan/RPL); Reduced Impact logging (RIL) and RIL-Carbon (RIL-C); Sustainable forest management (Sustainable Forest Management/SFM); Fire Awareness Community; Guarding against forest and land fires and illegal logging; Sustainable Forest Management certification and Forest Stewardship Council (FCS); Timber Legality Verification System (Sistem Verfifikasi Legalitas Kayu/SVLK |
| The regional apparatus organizations (RAOs) | 9 | Determination of Climate Villages; Fire Awareness Community; develop five schemes of Social Forestry; determination of disaster-resilient villages; new and renewable energy; building Village Index Development; Fire Awareness Farmers Group |
| Development partner | 14 | Sustainable forest management; community conservation area; sustainable energy; rhinoceros habitat protection and carbon calculation; SIGAP Village; sustainable oil palm plantation development policy; peat social forestry and sustainable peat management and protection; Green growth planning, FCPF-CF support, institutional capacity improvement, planning, and business plan development; Conservation, protection, restoration, and sustainability of forest use; sustainable community oil palm plantation development; Social forestry, conflict management; budget transparency; Forest carbon management; Community training in relation to carbon and conflict resolution; Out-of-forest forest planning and management; Local institutional improvement, mangrove forest restoration, and local conservation; Mangrove forest restoration |
| Field of Business | 6 | Fire Awareness Community Farmers Group; Sustainable Palm Oil Estate under Indonesian Sustainable Palm Oil Certification (ISPO) and Roundtable on Sustainable Palm Oil (RSPO; High Conservation Value (HCV) plantation; High Conservation Value Forest (HCVF); Corporate Social Responsibility (CSR); Post-mining reclamation; Environment impact assessment (Analisis Dampak Lingkungan/AMDAL); environment management plan (Rencana Pengelolaan Lingkungan/RKL); environment monitoring plan (Rencana Pemantauan Lingkungan/RPL); Reduced Impact logging (RIL) and RIL-Carbon (RIL-C); Sustainable forest management (Sustainable Forest Management/SFM); Fire Awareness Community; Guarding against forest and land fires and illegal logging; Sustainable Forest Management certification and Forest Stewardship Council (FCS); Timber Legality Verification System (Sistem Verfifikasi Legalitas Kayu/SVLK |

Source: authors' compilation.

### 3.1.3. Commitments of East Kalimantan Provincial Government as Sub-National Entity to Implement the CVP plus as Part of Climate Change Controlling Program

As one of the provinces with vast forest cover, the East Kalimantan Provincial Government has since 2008 been committed to reducing the rates of deforestation and forest degradation as well as to lowering the $CO_2$ emission rate by becoming part of the 10 provinces that initiated the establishment of the Governor's Climate and Forest Task Force. This commitment was followed by a declaration of "Kaltim Green" at the 1st East Kalimantan Summit and the readiness for REDD implementation in 2020, which aimed to improve the overall quality of life of the community, minimize ecological disasters, reduce pollution and degradation of water and air quality, and raise community awareness.

In 2011, the provincial government of East Kalimantan implemented this commitment by establishing the council on Climate Change of East Kalimantan (DDPI-Kaltim), which was tasked with formulating province-level strategies to reduce emissions and climate change mitigation. In the same year, the East Kalimantan Provincial Government made strategic documents such as the Low Carbon Growth Strategy (LCGS, document available), Regional Action Plan on Greenhouse Gas Emission Reduction (RAD-GRK) [27], the East Kalimantan Provincial Action Plan and Strategy (SRAP) on REDD+ Implementation [28], the Climate Change Master Plan [29], and the Green Economy Master Plan [30].

On the basis of this commitment, in 2015, the Minister of Environment and Forestry (MoEF) selected East Kalimantan Province as the location for the pilot project of Reduction of Emission from Deforestation and Forest Degradation under the result-based payment scheme through the World Bank-managed Forest Carbon Partnership Facilities-Carbon Fund (FCPF-CF) program. Under the FCPF-CF program, East Kalimantan Province will receive an incentive for every reduction in greenhouse gas emission in the land-based sector for the period 2020–2024. The East Kalimantan Provincial Government is set to integrate the FCPF-CF program into the Kaltim Green framework, affirming emission reduction for a longer period of time up to 2035 [30]. The CVP plus program is included within component 1 (Forest and Land Governance) and component 4 (Sustainable Alternative for Society) of the ERPD, and by the East Kalimantan Provincial Government, this program is to be incorporated in the Regional Medium-Term Development Plan (RPJMD) of East Kalimantan in 2019–2023 [31]. The planning of the CVP plus program through integration into the RPJMD of East Kalimantan is proof of the strong commitment of the East Kalimantan Provincial Government in efforts to control climate change. In the process, the integration planning requires the participation of all parties from the provincial level to the village government and village communities/customary communities. This participatory process is certainly not an easy thing because each party, especially the village/customary community, has different social, economic, and ecological characteristics. In addition, the main challenge is how to ensure that the RPJMD can actually be operationally translated to the village community/customary community level.

The implementation of the CVP plus in the FCPF Project requires intensive coordination and communication between the Provincial Government of East Kalimantan, the Regency Government to the village government/customary community level. This is a challenge that needs to be handled properly in the hierarchical management of activities from the top level to the bottom level. In FCPF-CF implementation, a management unit has been founded at the sub-national level to manage the FCPF-CF program under the Governor as the director, Regional Secretary as the technical commission chief, the Economy and Development Assistant in the Program Management Unit (PMU), the Economic Bureau as the secretariat coordinator, and four Work Groups in the Program Management Unit (PMU), namely the Measurement, Monitoring, and Reporting (MMR) Work Unit that is coordinated by the Environment Service of East Kalimantan Province (Dinas Lingkungan Hidup/DLH Prov Katim), the Safeguard Work Unit that is coordinated by the Forestry Service of East Kalimantan Province (Dinas Kehutanan/Dishut Prov Kaltim), the Planning and Budgeting Work Unit that is coordinated by the Regional Planning and Development Agency (Bappeda) of East Kalimantan and Regional Financial and Asset Management

Agency of East Kalimantan, and the Benefit Sharing Mechanism (BSM) Work Unit that is coordinated by the Economic Bureau of the East Kalimantan Provincial Government, in reference to the national institution of FCPF-CF management [32].

Figure 4 above shows that in addition to the Governor, the Provincial Secretary who chaired the FCPF-CF activities at the sub-national level also reports the results of activities to the National Secretariat of REDD and the Ministry of Environment and Forestry. The Governor also coordinates with the National Steering Committee led by the Secretary-General of the Ministry of Environment and Forestry, which plays a role in preparing directions to the national technical committee and the National REDD+ Secretariat of the Ministry of Environment and Forestry, who then reports to donors. To support the responsibilities of the Governor, the sub-national institution of the FCPF management unit (Sub-National PMU), chaired by the Economics and Development Administration Assistant, provides facilitation of management and technical aspects.

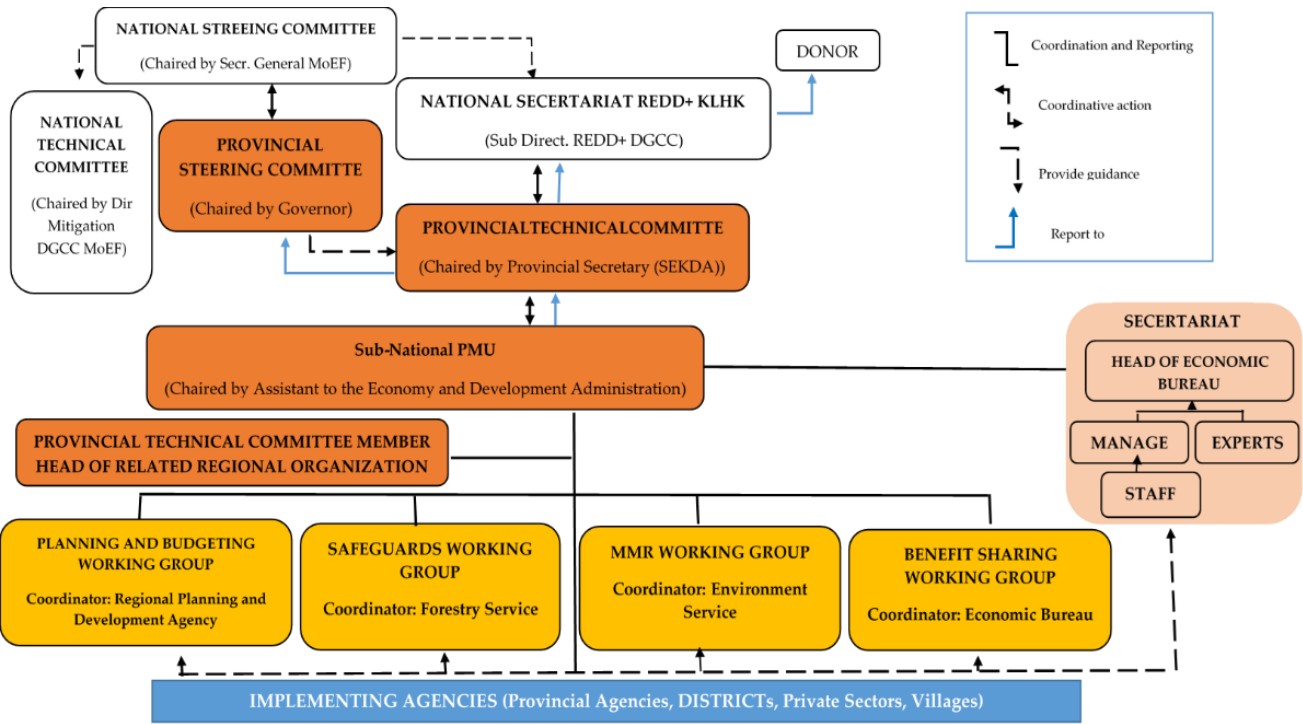

**Figure 4.** FCPF management unit structure at sub-national (East Kalimantan Province).

## 4. Discussion

### 4.1. The Implementation of the CVP plus in East Kalimantan Province

To date, the implementation of the CVP plus in East Kalimantan Province has achieved the FPIC implementation at the village level and received approval from 99 villages. The COVID-19 pandemic has hindered FPIC implementation in two regencies, namely Kutai Kertanegara Regency and Mahakam Ulu Regency. However, as a whole, the CVP plus implementation has proceeded as planned and will soon enter the phase where the village communities' emission reduction activities will be reported to the MMR portal of East Kalimantan, which is integrated with the Climate Change Control National Registration System operated by the Ministry of Environment and Forestry of the Republic of Indonesia.

The description in the results section proves that village communities in the 150 CVP plus villages in East Kalimantan Province are generally able to adapt to and mitigate climate change at the local level. This supports the rationale that village communities that are able to conserve forest cover will potentially gain incentives from FCPF-CF through the CVP plus program. This is also proof of the recognition that the Adat/Custom communities are the landowners [33,34]. Therefore, as a token of appreciation, the village communities in the

150 CVP plus villages receive the largest proportion of incentives from performance (65%) and rewards (10%) if the villages succeed in preventing deforestation and land degradation within a 10 year period from 2006 to 2016 [1,35].

### 4.2. Lessons Learnt from the CVP plus Implementation in East Kalimantan

4.2.1. Preparation of 150 Villages Determination

In conducting the CVP plus, the FPIC team performed socialization at the province, regency, and village levels according to the FPIC guide [13,15,16,18,31,32]. At the province level, the socialization covered the introduction to the program as well as the scope, stages, and follow-up plan of the program. At the regency level, the socialization was intended to ensure role distribution, willingness, and ways of reaching the villages. Meanwhile, at the village level, the socialization was undertaken in two steps: (1) ensuring that the community members understood the program, demonstrated no objection, and expressed their agreement, and (2) ensuring that the community members understood their rights and obligations and ensuring their commitment (see Figure 5).

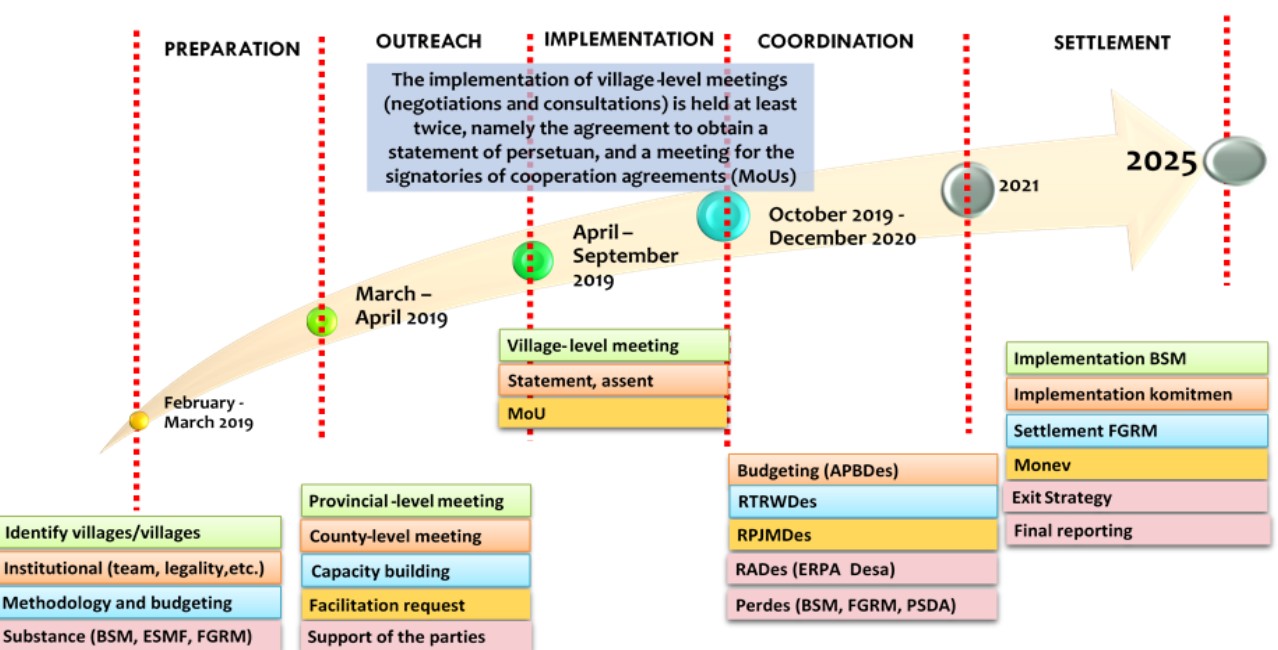

**Figure 5.** Stages of the CVP plus Implementation in East Kalimantan.

The 150 CVP plus villages were spread across 8 cities/regencies, had variations in livelihoods, and had Adat/Custom and newcomer communities (see Table 2).

The forests present in the 150 CVP plus villages were highland natural forests, lowland natural forests, peat swamp forests, and mangrove forests with various areas of forest cover and carbon stocks, with the lowest in Balikpapan city and the highest in Berau Regency and Mahakam Ulu Regency (see Table 3). The villages selected for the CVP plus program are villages with good forest cover. Good forest cover in the CVP plus has a high carbon stock, which contributes significantly to preventing emissions from deforestation and forest degradation. Therefore, good forest cover and high carbon stocks are very important for the CVP plus program.

**Table 2.** Distribution of 150 CVP plus villages by regency, livelihood, the existence of Adat/Custom communities, and the existence of newcomers.

| City/Regency | Number of Villages | Livelihood | Existence of Adat/Custom Communities | Existence of Newcomers |
|---|---|---|---|---|
| Balikpapan City | 2 | Private employees, farmers, laborers, and professionals | Existant but unacknowledged | Existant |
| Paser Regency | 19 | Swidden farmers, plantation farmers, forest gatherers, and private employees | Existant and acknowledged | Non-existant |
| Penajam Paser Utara Regency | 3 | Fishermen, Swidden farmers, and private employees | Existant but unacknowledged | Existant |
| Kutai Kartanegara Regency | 24 | Swidden farmers, fishermen, plantation farmers, private employees, and forest gatherers | Existant but unacknowledged | Existant |
| Kutai Timur Regency | 19 | Swidden farmers, plantation farmers, forest gatherers, and private employees | Existant but unacknowledged | Existant |
| Kutai Barat Regency | 22 | Swidden farmers, plantation farmers, forest gatherers, private employees, and fishermen | Existant and acknowledged | Non-existant |
| Berau Regency | 38 | Swidden farmers, forest gatherers, plantation farmers, fishermen, and private employees | Existant but unacknowledged | Existant |
| Mahakam Ulu Regency | 23 | Swidden farmers, plantation farmers, forest gatherers, fishermen, and swallow farmers | Existant and acknowledged | Non-existant |
| Total | 150 | | | |

Source: authors' compilation.

**Table 3.** Forest cover and carbon stock potential in the CVP plus villages per city/regency.

| City/Regency | Forest Type | Number of Villages | Forest Cover Area (ha) | Carbon Stock (Mt) | Proportion of C-Stock (%) |
|---|---|---|---|---|---|
| Balikpapan City | Mangrove forest | 2 | 8450 | 0.002 | 0.15 |
| Paser Regency | Highland natural forest, mangrove forest | 19 | 358,806 | 0.054 | 5.39 |
| Penajam Paser Utara Regency | Lowland natural forest, mangrove forest | 3 | 23,546 | 0.003 | 0.34 |
| Kutai Kartanegara Regency | Highland natural forest, peat swamp forest, mangrove forest | 24 | 707,901 | 0.148 | 14.78 |
| Kutai Timur Regency | Highland natural forest, peat swamp forest | 19 | 826,967 | 0.154 | 15.42 |
| Kutai Barat Regency | Lowland natural forest, highland natural forest, peat swamp forest | 22 | 390,153 | 0.058 | 5.83 |
| Berau Regency | Highland natural forest, lowland natural forest, mangrove forest | 38 | 1,237,928 | 0.243 | 24.32 |
| Mahakam Ulu Regency | Highland natural forest, lowland natural forest | 23 | 1,495,791 | 0.338 | 33.76 |
| Total | | 150 | 5,049,541 | 1.001 | 100.00 |

Source: Land Cover Map of East Kalimantan in 2019 [36] analyzed.

Forest carbon stock of the 0–20 Mt class was distributed across 8 regencies, accounting for 673.2 Mt. In class 20–40 Mt, there was Mahakam Ulu Regency with a total of 121.6 Mt, and in class 40–60 Mt, there were 4 regencies, namely Kutai Kertanegara Regency, Kutai

Timur Regency, Berau Regency, and Mahakam Ulu Regency, with a total of 205.9 Mt. See Figures 6 and 7 for more detail.

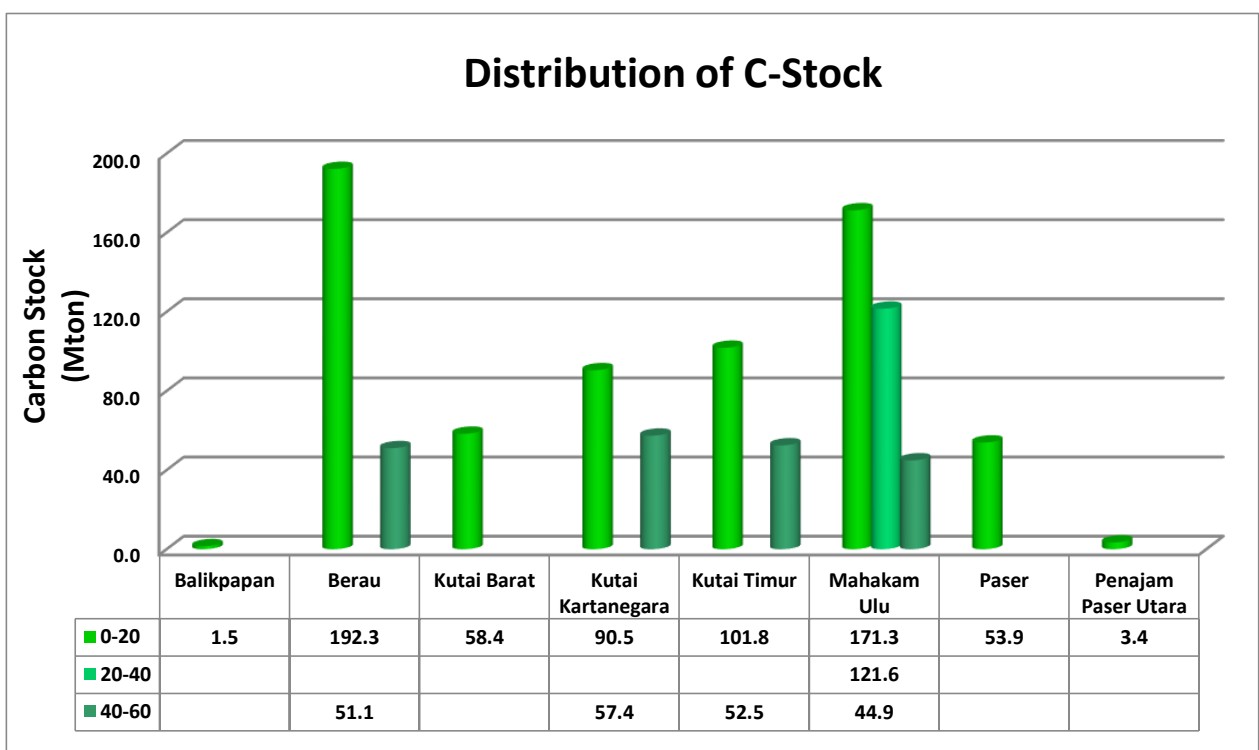

| | Balikpapan | Berau | Kutai Barat | Kutai Kartanegara | Kutai Timur | Mahakam Ulu | Paser | Penajam Paser Utara |
|---|---|---|---|---|---|---|---|---|
| ■ 0-20 | 1.5 | 192.3 | 58.4 | 90.5 | 101.8 | 171.3 | 53.9 | 3.4 |
| ■ 20-40 | | | | | | 121.6 | | |
| ■ 40-60 | | 51.1 | | 57.4 | 52.5 | 44.9 | | |

**Figure 6.** Carbon stock distribution across 150 CVP plus villages in 8 regencies/cities.

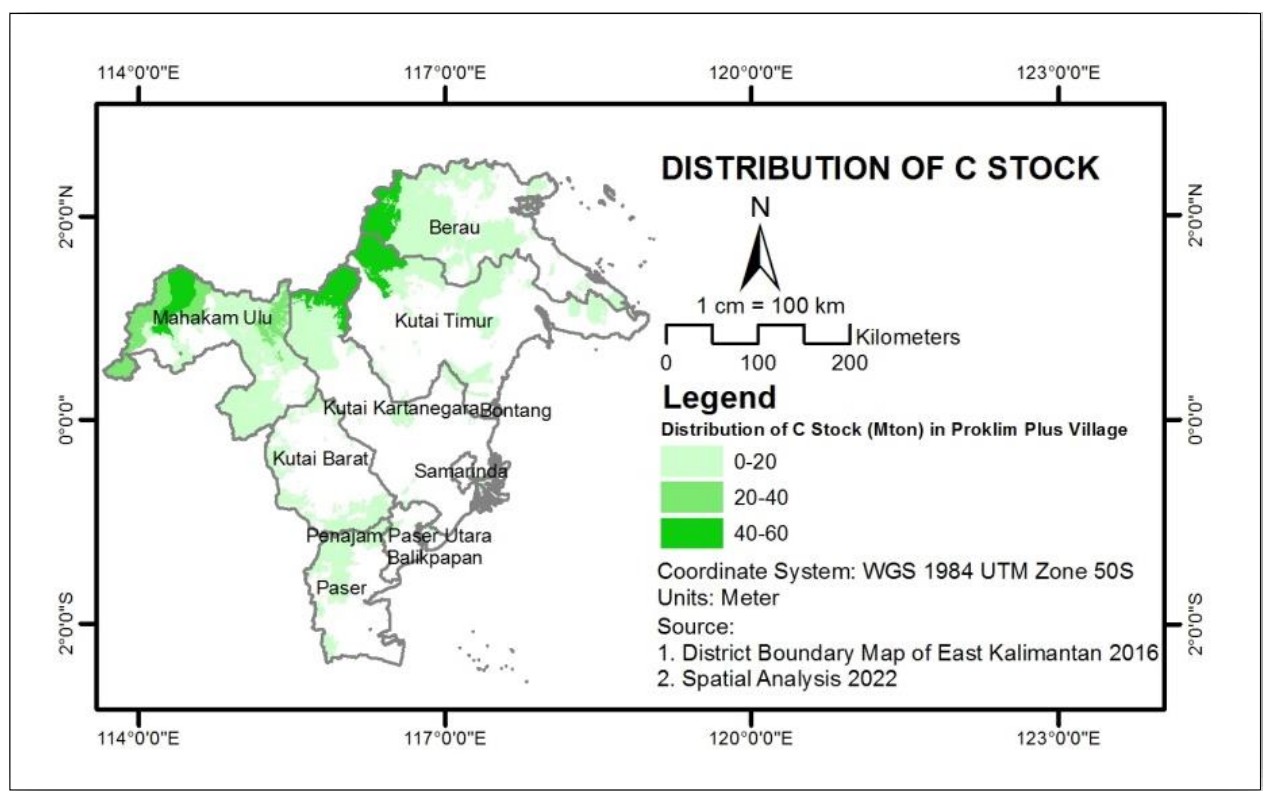

**Figure 7.** C-stock distribution in East Kalimantan Province. Source: author's compilation.

The existence of Adat/Custom communities constitutes one of the prominent criteria in determining the 150 villages to be targeted for the CVP plus. This is in line with some reports [14,17,37,38] that state that Adat/Custom communities' participation in the REDD+ activity is critical as it is related to the declaration of human rights. The existence of Adat/Custom communities in East Kalimantan Province is regulated by Governor Regulation No. 1 of 2015 on the Guide to the Acknowledgment and Protection of Adat/Custom Communities in East Kalimantan Province [39]. The East Kalimantan Provincial Government will acknowledge the existence of an Adat/Custom community if the following 5 requirements are met: there is a history of migration; there is an Adat/Custom area; there is an Adat/Custom institution; there is an Adat/Custom law, and there is an Adat/Custom object and site evidence of Adat/Custom community heritage.

Of the eight regencies/cities in which the FCPF-CP program was conducted, only three regencies had acknowledged the Adat/Custom communities, as evidenced by the existence of a Regional Regulation [39]. The three regencies/cities are Kutai Barat [33], Mahakam Ulu [34], and Paser [40]. Therefore, the determination of the 150 CVP plus villages was based on the representativeness of the native tribes and newcomers in East Kalimantan Province. Therefore, the determination of 150 CVP plus villages in East Kalimantan Province needs to take into account the representation of indigenous peoples and migrants.

4.2.2. Lessons Learnt from the 10 CVP plus Villages in East Kalimantan

The determination of the 10 CVP plus villages was based on the forest type, Adat/Custom community existence, ability to mitigate and adapt to climate change sustainably, and community livelihoods. The administrative and socio-economic condition data of the 10 CVP plus villages can be seen in Table 4.

**Table 4.** The administrative and socio-economic conditions of the 10 CVP plus villages.

| Regency/Village | Area (km²) * | Population Size (people) * | Number of Families (Families) * | Building Village Index Status ** | Access to the Village *** |
|---|---|---|---|---|---|
| | | | Kutai Barat Regency | | |
| Sembuan Village | 388.51 | 560 | 276 | Developed village | By rivers and land roads |
| Bermai Village | 39.56 | 299 | 97 | Underdeveloped village | By rivers and land roads |
| | | | Kutai Timur Regency | | |
| Senyiur Village | 122.395 | 4645 | 1324 | Underdeveloped village | By rivers and land roads |
| Melan Village | 447.48 | 563 | 169 | Developing village | By land roads |
| | | | Berau Regency | | |
| Merasa Village | 345.99 | 1170 | 265 | Developed village | By land roads |
| Long Lanuk Village | 427.11 | 887 | 254 | Developing village | By land roads |
| Punan Segah Village | 393.4 | 84 | 30 | Developing village | By rivers and land roads |
| Pegat Batumbuk Village | 547.18 | 858 | 236 | Developing village | By rivers and land roads |
| | | | Paser Regency | | |
| Swan Slutung Village | 495.78 | 633 | 204 | Developing village | By land roads |
| Muara Andeh Village | 148.24 | 289 | 139 | Underdeveloped village | By land roads |

Source: * Statistics Indonesia of East Kalimantan Province in Numbers (2021), ** Community and Village Empowerment Service of East Kalimantan Province (2021), and *** Authors' compilation.

The 10 CVP plus villages had a land cover area in the form of forests of 1113.18–49,134.87 ha and a carbon stock potential of 0.16–8.02 Mt (see Table 5).

In general, the villagers' livelihoods were strongly related to the natural resources potential existing around the forest, such as timber and non-timber forest products such as rattan, honey, medicinal herbs, waterfall, game, and fish. Additionally, according to the results of a study in the villages along the Ratah River of East Kalimantan Province [41] their livelihoods were also influenced by existing companies, such as those running the palm oil plantations, timber, and coal mining fields. The social, economic, and cultural conditions of the 150 CVP plus villages can be seen in Table 6.

**Table 5.** Forest cover condition and carbon stock potential of the 10 CVP plus villages.

| Regency/Village | Forest Cover Area (ha) | Potential Carbon Stock (Mt C) | Forest Type |
|---|---|---|---|
| Kutai Barat Regency | | | |
| Sembuan Village | 27,089 | 4.08 | Lowland natural forest |
| Bermai Village | 13,473 | 1.98 | Lowland natural forest |
| Kutai Timur Regency | | | |
| Senyiur Village | 19,116 | 3.45 | Peat swamp forest |
| Melan Village | 1113 | 0.16 | Highland natural forest |
| Berau Regency | | | |
| Merasa Village | 22,544 | 4.42 | Highland natural forest |
| Long Lanuk Village | 23,373 | 3.62 | Highland natural forest |
| Punan Segah Village | 35,088 | 8.02 | Lowland natural forest |
| Pegat Batumbuk Village | 16,540 | 2.82 | Mangrove forest |
| Paser Regency | | | |
| Swan Slutung Village | 49,135 | 7.95 | Highland natural forest |
| Muara Andeh Village | 9545 | 1.41 | Lowland natural forest |

Source: Land Cover Map of East Kalimantan in the Year 2019 [36] analyzed.

**Table 6.** Social, economic, and cultural conditions as well as natural resource potential.

| Regency/City | Occupation | Dominant Tribe | Natural Resources Potential | Companies Existing around the Village |
|---|---|---|---|---|
| | | Kutai Barat Regency | | |
| Sembuan Village | Swidden farmers, plantation farmers, non-timber forest product gatherers | Dayak Benuaq Tuayaatn | Timber, rattan, non-timber forest products | Timber company |
| Bermai Village | Swidden farmers, palm oil plantation company employees, coal mining company employees | Dayak Benuaq Indaatn | Timber, rattan, nature tourism, palm oil | Palm oil company, coal mining company |
| | | Kutai Timur Regency | | |
| Senyiur Village | Swidden farmers, fishermen | Kutai Senyiur | Palm oil plantation, swamp/peat timber, fish | Palm oil company, timber company, coal mining company |
| Melan Village | Swidden farmers, palm oil plantation company employees | Dayak Bahau Modang Long Way | Rubber plantation, palm oil, orchard, bamboo forest, aren palm plantation | Palm oil plantation company |
| | | Berau Regency | | |
| Merasa Village | Swidden farmers, palm oil, rubber, and cacao farmers, duck, cow, and goat farmers, vegetable farmers | Dayak Kenyah Uma' Bakaq, Kayan | Nature tourism, forest, and non-timber forest products, palm oil plantation, rubber, orchard | Coal mining company, timber company |
| Long Lanuk Village | Swidden farmers, coal mining company employees, civil servants | Dayak Ga'ai, Dayak Kenyah Uma' Kulit | Animal, honey, timber, swift's nest, agarwood, rattan | Coal mining company |
| Punan Segah Village | Swidden farmers, gold miners, hunters, fishermen, non-timber forest product gatherers, pig and chicken farmers | Dayak Punan | non-timber forest products, animals, fish, gold | Timber company |

**Table 6.** *Cont.*

| Regency/City | Occupation | Dominant Tribe | Natural Resources Potential | Companies Existing around the Village |
|---|---|---|---|---|
| Pegat Batumbuk Village | Fishing fishermen, pond farmers, entrepreneurs, lading farmers | Bugis-Makasar | Mangrove timber, prawn seeds, crabs, fish, prawns | Coal mining company |
| | | Paser Regency | | |
| Swan Slutung Village | Swidden farmers, hunters, non-timber forest product gatherers, plantation farmers | Paser, Muluy | Natural forest, timber, animal, honey | Timber company |
| Muara Andeh Village | Swidden farmers, hunters, plantation farmers, honey gatherers | Dayak Paring Sumpit | Natural forest, honey, fruits, timber, clean water | Timber company, energy company |

Source: authors' compilation.

In essence, natural resources management and utilization by communities are customs and local wisdom that are passed down from generation to generation. Therefore, they are performed sustainably according to the prevailing Adat/Custom rules, either written or unwritten. Sustainable forest resource management and utilization receive support from a variety of parties through managing institutions, as can be seen in Table 7.

**Table 7.** Natural resources management and utilization.

| Regency/Village | Natural Resources Management and Utilization Pattern | Sustainable Forest Resources Management Efforts | Managing Institutions | Supporting Institutions |
|---|---|---|---|---|
| | | Kutai Barat Regency | | |
| Sembuan Village | Rattan farming (*simpukng We'*), Fruit farming (*simpukng bua'*) | Non-timber forest products utilization under the Village Forest and Adat/Customary Forest schemes | Sebuan Village Forest Management Institution, Jumetn Tuwoyatn Adat/Custom Institution to manage Datai Nirui Adat/Customary Forest | KPHP (Production Forest Management Unit) Damai, Aliansi Masyarakat Adat/Custom Nusantara (AMAN) Kaltim |
| Bermai Village | Rattan farming (*simpukng we'*), Fruit farming (*simpukng ruyan/lembo*) | Non-timber forest products utilization under the Village Forest scheme | Bermai Village Forest Management Institution, Adat/Custom Institution | KPHP Production Forest Management Unit) Damai |
| | | Kutai Timur Regency | | |
| Senyiur Village | Land utilization as palm oil plantation, peat forest, and *kenohan*/lake for fishing | Swamp and peat conservation, swamp and peat restoration in Loah Putih, fish protection at Suwi Lake, *bekantan* protection along Senyiur River, crocodile protection at Suwi Lake | Mesangat Suwi Wetland Essential Ecosystem Estate Forum | Konsorsium Yayasan Konservasi Katulistiwa Indonesia (Yasiwa)-Ulin Foundation |
| Melan Village | Land use as palm oil plantation, rubber plantation, orchard, nature and cultural tourism, agarwood farming | Spring conservation, Dengkelih Heritage Forest protection in Dusun Ngelin | Dengkelih Heritage Family, Melan Village Adat/Custom Institution | Yasiwa-Ulin Foundation |
| | | Berau Regency | | |
| Merasa Village | Land use as palm oil plantation, rubber plantation, orchard, nature and cultural tourism, agarwood farming | Tourism village management (natural and cultural tourism) | Merasa Village Government | KPHP (Production Forest Management Unit) Berau Barat Center Orangutan Protection (COP), GIZ Forclime FC, |

**Table 7.** *Cont.*

| Regency/Village | Natural Resources Management and Utilization Pattern | Sustainable Forest Resources Management Efforts | Managing Institutions | Supporting Institutions |
|---|---|---|---|---|
| Long Lanuk Village | Non-timber forest products utilization (animal hunting, honey, timber, rattan, agarwood, swallow) utilization, orchard farming (*stangu*) | Long Lanuk Village forest management, agroforestry plot development through Forestry Partnership | Long LanukVillage Forest Management Institution, Leboq Nyapa Indah Forest Farmers Group | KPHP (Production Forest Management Unit) Berau Tengah, Dipterocarps Research and Development Center, GIZ Forclime FC |
| Punan Segah Village | Non-timber forest products utilization (animal hunting, fish, gold), fruit tree planting, medicinal herb planting, *kelulut* farming | Punan Segah Village Forest management | Punan Segah Village Forest Management Institution | KPHP (Production Forest Management Unit) Berau Barat, GIZ Forclime FC |
| Pegat Batumbuk Village | Mangrove forest utilization for non-timber forest products (mangrove timber, mangrove fruits, fish, prawn seeds, crabs, and prawns) | Village forest management for social forestry, Batumbuk Watershed rehabilitation | Pegat Batumbuk Village Forest Management Institution | KPHP (Production Forest Management Unit) Berau Utara, Mulawarman Environment Forum (Forum Lingkungan Mulawarman/FLiM) |
| | | Paser Regency | | |
| Swan Slutung Village | Non-timber forest products utilization (timber, animal hunting, honey) and for shifting cultivation (*umo*), rattan farming (*sipukng rotan*), fruit farming (*sipukng buah*) | Communal land conservation, consisting of village, hunting ground, Adat/Customary forest, and field farmers | Kampong Mului Adat/Custom Institution | Padi Foundation |
| Muara Andeh Village | Non-timber forest products utilization (honey (*tanyut*), fruits, timber, clean water), orchard management (*sipukng*) | Forest (*alas/katuan*) conservation, communal land conservation for hunting ground, bamboo forest conservation, and Adat/Customary forest protection | Dayak Paring Sumpit Adat/Custom Institution | Padi Foundation |

Source: authors' compilation.

Community activities are also related to the programs offered by the village through the village institution or Adat/Custom institution. Work programs may come from partners, in this case, the Central Government/Ministry of Environment and Forestry such as the Center for Social Forestry and Environment Partnership of Kalimantan Area, Dipterocarps Research and Development Center, East Kalimantan Provincial Government such as Forestry Service of East Kalimantan Province and Forest Management Unit, and development partners such as (Yayasan Konservasi Alam Nusantara/YKAN, Deutsche Gesellschaft für Internationale Zusammenarbeit (GIZ) Forclime, GIZ Propeat, World Wide Fund for Nature (WWF) Indonesia, etc.). Some of the activities performed are actually purely initiated by the communities; for instance, the proposed activities by the Adat/Custom communities in Sembuan Village, Swan Selutung Village, and Muara Andeh Village under much facilitation from Aliansi Masyarakat Adat/Custom Nusantara (AMAN) of East Kalimantan and Yayasan Padi Indonesia.

There are also forest protection and management activities, such as one funded by TFCA Kalimantan in the Mesangat Suwi Wetland Ecosystem Estate (KEE LBMS) of Senyiur Village and Melan Village of Kutai Timur Regency under the facilitation of the YASIWA-ULIN Foundation Consortium. The Village Forest scheme is a program by the Directorate General of Social Forestry and Environment Partnership of the Ministry of Environment and Forestry in the Social Forestry Division that is implemented by Sembuan Village, Bermai Village, Long Lanuk Village, Punan Segah Village, and Pegat Batumbuk Village to give access for the communities to the forest and improve their economic conditions through Social Forestry Enterprise Group (KUPS) under the facilitation of the Kawal Borneo Community Foundation (KBCF)-Komunitas Konservasi Indonesia and Warung Informasi Konservasi (KKI-WARSI) of Jambi-Deutsche Gesellschaft für Internationale

Zusammenarbeit (GIZ) Forclime consortium, which later became part of the work of the Forest Management Unit.

### 4.3. The Challenges in the CVP plus Implementation

The challenge faced by the East Kalimantan Provincial Government in the CVP plus implementation at the village level is the low capacity of village human resources. The Local Government and Development Partners play a critical role in supporting village communities to integrate the CVP plus into the Village Medium-Term Development Plan (Rencana Pembangunan Jangka Menengah Desa/RPJMDes), the Village Budget (Anggaran Pendapatan dan Belanja Desa/APBDes), and the Village Spatial Plan (Rencana Tata Ruang Wilayah Desa/RTRWDes [42]. This integration process of the CVP to the Village Medium-Term Development Plan should ensure harmony between human well-being and ecology. In addition, the village communities are also in need of assistance in making a report of the emission reduction activity they have conducted on the MMR portal. The Community and Village Empowerment Service of East Kalimantan Province is a regional apparatus organization in East Kalimantan responsible for village-level assistance. In order to integrate the emission reduction program into the village development plan, the Community and Village Empowerment Service of East Kalimantan Province has proposed a green assistance program. Human Resource capacity improvement is also carried out by the development partners who have a work program in the same village.

### 4.4. The Strategies in the CVP plus Implementation

The success of the CVP plus implementation in East Kalimantan Province is the result of the collaboration and active role of all stakeholders, whether they be the central government, local government, development partners/non-governmental organizations, private parties, or communities. Multi-stakeholder partnerships have the potential to overcome the limitations in program implementation in relation to climate change [43]. Multi-stakeholder partnerships are a form of governance that can harness the strengths of different parties and actors. The climate change program is characterized by undesirable complexity and the need for policy coordination vertically, horizontally, and across sectors, including non-governmental organizations (NGOs), industry associations, and different levels of government (national and sub-national) [44,45]. The success of a program is highly reliant on the participation of all parties across multiple levels [46,47]. A social learning process in East Kalimantan for the implementation of the CVP is affected by the involvement of all actors/stakeholders and is very important to develop the existing social networks at the village level. This was the case in Central Java Province, in which 17 actors from the government, private sector, community and community groups, non-governmental organizations, mass media, and public legal entities were involved [48]. The collaborative governance involving companies, the government, and the community constitutes a key to the success of the climate change program in Talanbubuk Village in Palembang, South Sumatera Province [49]. Community empowerment and multiple stakeholders' involvement seem to characterize this joint adaptation-mitigation program. In addition, the joint adaptation-mitigation program in the CVP plus allows for better measurement and verification as well as monitoring and evaluation under the forest cover criteria around the village area. Regular program monitoring and evaluation is necessary to ensure all activities are on the right track.

Approval from the village communities as to the CVP plus implementation through FPIC activity will be required. The same process also took place in the REDD+ activity in Africa [50] and Vietnam [38]. The CVP plus must be part of the village development plan that is set out in the Village Medium-Term Development Plan and the Village Budget. Furthermore, it must also be incorporated into the Village Spatial Plan and reinforced by a Village Regulation. The CVP plus should also integrate the Village Medium-Term Development Plan into the emission reduction program that contributes to the non-carbon benefits related to better forest management and livelihoods, especially the improvement

of local communities' access to forest resources for improved livelihoods. The program integration into the sub-national development planning and its synergy among stakeholders are needed to ensure optimal results [51].

The CVP plus implementation in East Kalimantan Province not only supports the CVP implementation but also supports the Social Forestry (SF) program of the Ministry of Environment and Forestry. SF is a program that provides the local communities with access to manage forest resources to improve their welfare as well as to resolve conflicts under five schemes: Village Forest, Community Forest, People's Crop Forest, Forestry Partnership, and Adat/Custom Forest. The CVP plus support in the SF program is included in Component 4 of the FCPF-CF, that is, a sustainable alternative for the community.

### 4.5. The Key Findings in Assessing the Challenges and Strategies of the CVP plus

The key findings in assessing the challenges and strategies of the CVP plus in East Kalimantan can be seen in Table 8.

**Table 8.** The key findings in assessing the challenges and strategies of the CVP plus in East Kalimantan.

| Challenges | Sustainable Strategies | | |
| | Strategy 1 | Strategy 2 | Strategy 3 |
| --- | --- | --- | --- |
| For the institutional aspect, there is a wide range of parties (communities, government, organizations, and development partners) that can be involved in a policy of CVP plus implementation, depending on the level of enactment (from local to national) and the type of policy (from regulation to statute). | Multi stakeholders can help to identify and intensively coordinate resources and support. For example, one of the development partners may coordinate or develop education and communication related to CVP plus activities. | Multi stakeholders can help to provide support for large-scale changes to existing processes of the CVP plus implementation. For example, a partner might help set up a website with information and implementation guidance. | Multi stakeholders can help all villages to establish institutional mechanisms and processes, such as a CVP activities observatory, working groups, and/or CVP co-ordination committees in order to expand the evidence and to promote policy dialogue on CVP issues as well as holding all actors accountable concerning CVP policies and actions initiated. |
| For integration into the development plan, the CVP activity has not been included in the Village Medium-Term Development Plan. It is important to acknowledge some of the limitations of this analysis by contextualizing the findings: data collection took place three years after launching the CVP plus, while inter-sectoral CVP plus policy implementation and its development require time. There is a lack of capacity and low awareness of the human resources in the implementation of CVP at the village level. This review provided the reference against which, subsequently, the implementation status of the commitments, where available, could be verified. | The CVP plus of villages have set goals for their activities, but a timeframe and related milestones were missing for most commitments. This affected the possibility of systematically assessing and comparing progress in the CVP commitment implementation. | The 150 CVP plus villages are firm in their commitment and primarily focused on implementing the CVP plus policy to encourage retention in rural environments with diverse natural resource ecosystems. | The 150 CVP plus villages should receive support in investing and managing the development of village community activities and building capacity in management, monitoring, and governance. |
| | Development partners (DPs) can be instrumental in developing and implementing the CVP actions in partnership with communities and village governments. | A number of organizations or development partners (DPs) can support, develop (and want to harmonize) the CVP program. The work of these DPs or NGOs is often complementary to that of the government when it comes to training for villagers and village governments. | The 150 CVP plus villages addressed the pathway by diversifying the environment, education, and recruitment of different cadres or volunteers and investing more in the CVP activities at the community and village level. |

**Table 8.** *Cont.*

| Challenges | Sustainable Strategies | | |
| --- | --- | --- | --- |
| | Strategy 1 | Strategy 2 | Strategy 3 |
| The findings and analysis from the CVP commitments implementation indicate that inter-sectoral action, dedicated political support, a partnership approach, and sustained funding are of crucial importance to further advance the CVP development agenda. | The partnerships developed by DPs will create funding and momentum to put in place innovative policies and solutions such as training modules for villagers and community empowerment, etc. | Multi stakeholders can support and plan for policy, programmatic, and fiscal sustainability for the CVP plus implementation. For example, a development partner might create a strategic plan that identifies where funding will come from once initial funds are exhausted. | Village governments can create CVP plus activities by allocating village funds and funds from the private sector in the form of CSR. |

## 5. Conclusions

The integration of the CVP into the sub-national development plan and its implementation in the village community will be sustainable if the social learning process among all stakeholders in East Kalimantan is excellent. According to the information gathered from all stakeholders in East Kalimantan, the challenges for the CVP implementation in the future are institutional aspect, integration aspect in the development plan, human resource capacity aspect, and funding aspect.

The main feature of the CVP is the presence of forest lands, which are still in very good condition, with a total area of 5,049,541 ha and a total carbon stock of 1.001 gigatons. Mahakam Ulu Regency is very dominant, with a forest area of 1,495,791 ha and a carbon stock of 33.76% compared to the other regencies. The 10 CVP plus villages had a forest area of 1113–49,135 ha and a carbon stock potential of 0.16–8.02 Mt.

As for the institutional aspect, the CVP working group operates only at the provincial level, and there is neither a district-level working group nor an official institution charged with supporting the CVP after its establishment. Regarding the aspect of integration into the development plan, the CVP activity has not been included in the Village Medium-Term Development Plan because the meeting of all stakeholders has not yet been completed. Currently, the CVP is a priority program for the East Kalimantan Provincial Government. In the meanwhile, there is a lack of capacity and people's awareness to implement the CVP at the village level. In addition, the government's budget for the CVP implementation is limited and needs cooperation and additional funding from all stakeholders (domestic and international).

To address the above challenges, the following strategies should be considered for future CVP implementation. Institutional strengthening is necessary to support and implement the CVP program, mainly at the village level *(desa* and *kelurahan)*. As part of the program planning, the integration of the CVP into the medium-term village development plan should consider environmental sustainability and economic aspects to ensure harmony between human well-being and ecology. The biophysical conditions of the village area, the characteristics of the village area, and the use of available space in the village should be identified to ensure healthy availability of water, food, and clothing, renewable energy sources, and the environment. Regarding the capacity of human resources, the activities of training, lessons learned, field visits, and workshops can be used to improve the capacity of human resources in terms of technical information, action steps, and work procedures, especially at the village level *(desa* and *kelurahan)*. In addition, collaboration between NGOs and other parties involved in climate change, communities, local government, and central government is needed, and funding opportunities from the private sector or international donors should be pursued. Collective actions and strong support from all stakeholders are very important to expand existing social networks at the village level and increase community resilience to address climate change impacts. Regular monitoring and evaluation of

the programs are very important for a long-term strategy. All stakeholders should take all steps of the monitoring and evaluation process to ensure that the CVP is on the right track.

**Author Contributions:** C.B.W. and E.M.A. performed the original draft. I.W.S.D. and C.K. conducted supervision. N.S., S.E. and T.W. performed the methodology of the manuscript. R.M. and Y.H. conducted review and editing. Y.N. and W.I.S. coordinated, integrated, and consolidated all resources. N.N. and A.A. performed validation of data and information. K.K. and A.N.L. performed visualization data and information. All authors have read and agreed to the published version of the manuscript.

**Funding:** This research received no external funding.

**Institutional Review Board Statement:** The research of our manuscript was privately funded, and received no external funding. All data and information were delivered according to the private research of each author. There is no conflict in our manuscript. All subjects gave their informed consent for inclusion before they participated in the study and in accordance with the Declaration of Helsinki.

**Informed Consent Statement:** Informed consent was obtained from all subjects involved in the study.

**Data Availability Statement:** The data set used/and/or analyzed during the current study are available from the corresponding authors on reasonable request.

**Acknowledgments:** We thank the kind support from the Ministry of Environment and Forestry, Republic of Indonesia; East Kalimantan Government; Development Partners; Regional Council of Climate Change of East Kalimantan Province; Forest Carbon Partnership Facility-Carbon Fund Program; Village Government and Community in 10 Selected Villages and anonymous reviewers for substantial and grammatical comments and corrections. This work was supported by the Ministry of Education, Youth and Sports of CR within the CzeCOS program, grant number LM2018123.

**Conflicts of Interest:** The authors declare no conflict of interest.

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
