# Peer review of "Challenges to and Strategies for the Climate Village Program Plus: A Lesson Learned from Indonesia"

_sustainability, doi:10.3390/su14095530_

Round 1

Reviewer 1 Report

This is a study that assess the challenges to and strategies for implementing the climate village program in the national and sub-national contexts, especially in Indonesia. The assessment results of the social learning of CVP Plus (those on the policy process and its concept, integration program and implementation) show that sustainable strategies need to be addressed  by integrating the CVP Plus into medium-term development plan of the region. Finally, the steps of implementation of CVP, development partners and their roles are discussed.

I think this study is a progress by deepening and expanding policy studies focused on CVP cases. Some revision suggestions are listed below:
1) Introduce the design and evolution of CVP program.
2) Discuss pros and cons of CVP program compared with other climate mitigation and resilience policies based on your research.

Author Response

Dear Reviewer 1,

Here, we attached our responds for your reviewed. 

  1. Respond of question 1 : Generally, we described it in the second paragraph (lines 42 – 47).  But, regarding your suggestion we added one more sentence about the new policy:

     “In 2021, the guidelines and direction for the implementation of CVP is issued by MoEF cq. Directorate of Climate Change” (lines 51 - 53)

  2. Respond of question 2 : Our manuscript does not describe about the prospects and consequences of implementing CVP Plus. However it describes more about the advantages and evolutions of CVP Plus compared to the other mitigation activities in Indonesia.

Thank you very much for your kind attention.

Best Regards,

Authors

Reviewer 2 Report

Challenges to and strategies for the climate village program plus : a lesson learned from Indonesia

Indonesia has recognized the need for all stakeholders and communities to work together to implement climate adaptation and mitigation actions, to make them more effective and wide reaching, and to achieve the national climate goals. The country also felt the needs for local communities to understand their climate vulnerabilities and be empowered to take informed actions to mitigate and adapt to these.

This article is based on Climate Village Program implemented in Kalimantan (Indonesia).  Climate Village Program (CVP) is one of the national flagship programs of the Indonesian Ministry of Environment and Forestry to support emission reduction and climate resilience since 2016. Thus, the community-based climate village program (ProKlim) has strengthened the implementation of integrated adaptation and mitigation impact of climate change, reduction of greenhouse gases emission, and recognition of active community participation. The paper summarizes the achievements and the challenges related to this CVP.

The objective of the authors is to assess the challenges and strategies in climate program implementation at the national and sub-national levels to support emission reduction and climate resilience, based on the CVP Plus lesson learnt in East Kalimantan. This is to be examine in line with the local development plan and other climate change programs with stakeholders, as envisioned in the regional medium-term development plan.

The methodology used help them to reach the expected results based on the climate village plus policy and program, the steps of its implementation, the role of the private corporation and the lesson learned. These lessons take three directions: Ecological/environmental (carbon stock distribution), socioeconomic and political (natural resources managements and utilization, natural resources potential) etc.

Really the text is well organized, but I have few remarks:

In page 9, I don’t see the interest of the figure 4 dealing with sub*national institution of FCPF management unit.

Table 9 in page 15 is too long and I advise the author to shorten it.

Thirdly, the conclusions need to provide some statistical results taken within the text (for example on carbon stocks etc.) 

Lastly I wonder myself if there is only one lesson or many lessons. I think there are many lessons. Idem for the conclusions are there many conclusions or a conclusion? I think it is a conclusion from paper.

Author Response

Dear Reviewer 2,

Here, we attached our responds of your reviewed.

1. Respond for question 1 : 

Actually, figure 4 illustrates that the implementation of FCPF program should be through a national program mechanism which is mandated to sub national/province. But to clarify our figure, we added this following sentences.

To support responsible of governor, then the sub-national institution of FCPF management unit (Sub-National PMU) that chaired by Economics and Development Administration Assistant provides facilitation of management and technical aspects (lines 298 - 301)

2. Responds for question 2 : We revised Table 9 according to your advised (lines 387 – 388)

3. Responds for question 3 : 

We agreed with your suggestion, and we added some sentences for clarified.

The main characteristic of CVP is existence of forest cover area that still very good condition with the total of forest cover area 5,049,540.86 ha with a total carbon stock 1.06 gigatonnes. Mahakam Ulu Regency is very dominant of forest cover area amounted 1,495,790.52 ha with carbon stock proportion as 32.94% compare to the others regencies. The 10 CVP Plus villages had a land cover area in the form of forest of 1,113.18–49,134.87 ha and carbon stock potential of 0.18–8.89 Mt (lines 501 -506)

4. Respond for question 4 : 

Yes, we agreed with your statement that our conclusion is a conclusion. It can be seen in our manuscript (lines 495 -500).

Meanwhile, other paragraph in conclusions are means to be detailed of a conclusion.

Thank you for your kind attention.

Best regards,

Authors

Reviewer 3 Report

I have not any special comments. The paper is of good quality.

Author Response

Dear Reviewer 3,

Thank you very much for your positive comments. 

Best Regards,

Authors

Reviewer 4 Report

This is a comprehensive study on the CVP Plus implementation although without any statistical analysis. However, I would suggest some points to improve this manuscript:

  1. Please elaborate on the Discussion part. Divide it into 4 - 5 subheadings that clearly stated your key findings. This is to make the readers easier to understand what is your key findings.
  2. The Discussion should clearly state the answers to your research questions as mentioned in the Introduction.
  3. Please construct a table to inform your readers about your key finding in assessing the challenges and strategies of the CVP Plus lesson learnt in East Kalimantan.
  4. Tables 1,2,3, should be joined to make them more concise.
  5. Part 3.2 Lesson learnt of CVP Plus Implementation in East Kalimantan should be in the discussion. This part should be reformated. Please keep data and information in the "Results" and move the analysis to the "Discussion".
  6. I don't understand the aims to present carbon stock calculation in this study. Please elaborate on this in the discussion. Do you mean that CVP Plus could increase C Stock? Then, please discuss this.
  7. I found it difficult to read some information in the figure because of the font quality, such as in Figures 4 and 5. Please reformat.
  8. Please do not claim that this study uses a Quantitative method unless the study conducts a statistical analysis. If the calculation of Carbon Stock is considered a quantitative approach, please explain it in more detail in the methodology. 
  9. I found some typos and grammatical errors. Please ask for help from a professional proof-reader to fix those problems.

Author Response

Dear Editor and Reviewers,

First, we would like to thank you for the insightful comments on our manuscript. We have analyzed and considered all comments in revising the manuscript and are now pleased to return it to you.
Below are detailed responses to each comment and how we addressed them.
We look forward to hearing from you.

Yours Sincerely,

The Authors

Here some responses of our manuscript :

  1. Question : Please elaborate on the Discussion part. Divide it into 4 - 5 subheadings that clearly stated your key findings. This is to make the readers easier to understand what your key findings is.  Respond : Thank you for your suggestions. The discussion section has been revised, elaborated and improved. In general, the section has been divided into 5 subheadings based on the recommendations of the reviewer, as follows: first in lines 301-320, the second in lines 321-440, the third in lines 441-458, the fourth in lines 459-502, and the fifth in lines 504-509.
  2. Question : The Discussion should clearly state the answers to your research questions as mentioned in the Introduction. Respond : Thank you very much for your comment. In general, more information has been added to respond the research questions in the discussion, especially in the second paragraph (lines 42 – 47), and the fifth paragraph (lines 69-79).  

  3. Question : Please construct a table to inform your readers about your key finding in assessing the challenges and strategies of the CVP Plus lesson learnt in East Kalimantan. Respond : We agree with your suggestion. We have made a table in lines 506-507.

  4. Question : Tables 1,2,3, should be joined to make them more concise. Respond : We agree with your suggestions. We have created it in lines 225-227.
  5. Question : Part 3.2 Lesson learnt of CVP Plus Implementation in East Kalimantan should be in the discussion. This part should be reformatted. Please keep data and information in the "Results" and move the analysis to the "Discussion". Respond : Thank you for your thoughts on this point. We have moved the Part 3.2 on Lesson learnt from implementation of CVP Plus Implementation in East Kalimantan to the discussion section.

  6. Question : I don't understand the aims to present carbon stock calculation in this study. Please elaborate on this in the discussion. Do you mean that CVP Plus could increase C Stock? Then, please discuss this. Respond : Thank you for your suggestions. We have added some explanation in lines 338-344: The forests present in the 150 CVP Plus villages were highland natural forests, lowland natural forests, peat swamp forest, and mangrove forests with various areas of forest cover and carbon stocks, with the lowest in Balikpapan city and the highest in Berau Regency and Mahakam Ulu Regency (see Table 3). The villages selected for the CVP Plus program are villages with good forest cover. Good forest cover in CVP Plus has a high carbon stock, which contributes significantly to preventing emissions from deforestation and forest degradation. Therefore, good forest cover and high carbon stocks are very important for the the CVP Plus program.

  7. Question : I found it difficult to read some information in the figure because of the font quality, such as in Figures 4 and 5. Please reformat. Respond : Thank you for your comment on this point. We have revised it and also reformatted it in lines 285-286 (Figure 4), and lines 330-331 (Figure 5).

  8. Question : Please do not claim that this study uses a Quantitative method unless the study conducts a statistical analysis. If the calculation of Carbon Stock is considered a quantitative approach, please explain it in more detail in the methodology. Respond : Thank you for your thoughts on this point. We have explained it in lines 148-150.

  9. Question : I found some typos and grammatical errors. Please ask for help from a professional proof-reader to fix those problems. Respond : Thank you for your suggestions. Finally, proofreading of the English language has been performed.

Round 2

Reviewer 4 Report

This manuscript has been improved